# METABALANCE: HIGH-PERFORMANCE NEURAL NETWORKS FOR CLASS-IMBALANCED DATA

## ABSTRACT

Class-imbalanced data, in which some classes contain far more samples than others, is ubiquitous in real-world applications. Standard techniques for handling class-imbalance usually work by training on a re-weighted loss or on re-balanced data. Unfortunately, training overparameterized neural networks on such objectives causes rapid memorization of minority class data. To avoid this trap, we harness meta-learning, which uses both an "outer-loop" and an "inner-loop" loss, each of which may be balanced using different strategies. We evaluate our method, MetaBalance, on image classification, credit-card fraud detection, loan default prediction, and facial recognition tasks with severely imbalanced data. We find that MetaBalance outperforms a wide array of popular strategies designed to handle class-imbalance, especially in scenarios with very few samples in minority classes.

## 1 INTRODUCTION

Deep learning researchers typically compare their classification models on standard benchmark tasks, but these well-curated datasets are overwhelmingly class-balanced (Krizhevsky et al., 2009; Russakovsky et al., 2015). In contrast, real-world datasets are often highly imbalanced. For example, in applications such as cancer detection or environmental disaster prediction, a disproportionate amount of data available comes from negative cases (Kail et al., 2020; Fotouhi et al., 2019) To make matters worse, methods designed for balanced datasets often fail on imbalanced ones.

When the training data is severely skewed, neural networks tend to memorize a small number of samples from minority classes and do not generalize at inference. This shortcoming is particularly harmful since the minority classes are often the ones we care most about. The most common methods to handle class imbalance work by re-sampling data to achieve better balance in the training set or by re-designing the loss to attend more strongly to minority class data. However, these methods do not prevent rapid overfitting to minority samples and are sub-optimal when applied to highly over-parameterized models that memorize data quickly.

We propose MetaBalance, a modification of the MAML meta-learning algorithm for training deep neural networks on class-imbalanced data. Intuitively, the proposed training scheme finds a set of network parameters such that, when fine-tuned for one iteration, the model achieves good performance on balanced data. We compare our algorithm to common techniques for class-imbalance and find that MetaBalance significantly outperforms existing methods on tasks with few samples in minority classes. In particular, we evaluate our method on image classification, credit-card fraud detection, loan default prediction, and facial recognition tasks. Finally, we explain why deep neural networks trained with MetaBalance achieve better generalization on class-imbalanced data.

## 2 BACKGROUND

Class imbalance is problematic because classes with poor representation may be ignored by a model at inference time. Consider, for example, a corpus of credit-card transactions in which only $1\%$ of transactions are fraudulent. In this case, a fraud classifier can trivially achieve $99\%$ accuracy simply by predicting that every transaction is non-fraudulent. When identifying fraudulent transactions is difficult, this is often the optimal strategy for a classifier to take. For this reason, data scientists often care about optimizing the performance on a more *balanced* dataset that does not promote this strong bias.

In this section, we review existing methods, both classical and recently proposed, for training in situations of extreme class imbalance. We also give a brief overview of the MAML algorithm as it forms the foundation for our method.

## 2.1 LEARNING ON CLASS IMBALANCED DATA

Existing methods for handling class imbalance can be broadly split into three groups: re-sampling methods that increase the numbers of minority class samples, methods for reducing majority class samples, and classifier-level methods that modify the training routine to shift the model's focus towards minority samples during training.

**Oversampling** methods focus on generating new minority class samples from the available unbalanced data. A naive approach is to simply replicate points from the minority class, however this does not produce new information about the minority class and is known to lead to severe overfitting to over-sampled examples. To address this problem, Chawla et al. (2002) propose SMOTE, which generates unique minority samples by linearly interpolating between the existing observations from minority classes. Several improvements to SMOTE have been made with the goal of generating additional training data in a way that results in better decision boundaries after training (Han et al., 2005; Nguyen et al., 2011; He et al., 2008). For example, SVMSmote generates the new minority examples along the boundary found by a support vector machine(Han et al., 2005).

SMOTE and its modifications are intended for tabular data and not for high-dimensional data such as images. However, some strong data augmentation techniques designed for preventing overfitting on images operate in similar ways. For example, mixup generates new images by taking a convex combination of images in the dataset (Zhang et al., 2017) and CutMix blends two images by cutting a patch from one image and inserting it into another (Yun et al., 2019). Both methods produce labels for the new samples by taking a weighted average of labels of the blended images. SMOTE is closely related to mixup, with the main difference being that SMOTE only performs mixing within the minority class, while mixup intermingles samples between all classes. Kim et al. (2020) propose translating majority samples to minority samples by applying adversarial perturbations and in this way leveraging the diversity of the majority information. Finally, there is a growing body of work proposing GANs for generating realistic samples from minority classes, but training GANs is difficult, and these models are notorious for performing poorly on diverse datasets or memorizing their training data (Shamsolmoali et al., 2020; Deepshikha & Naman, 2020; Ali-Gombe & Elyan, 2019; Mullick et al., 2019).

**Undersampling** is another common technique for handling class-imbalance. In contrast to oversampling, which adds minority class data, undersampling removes from the majority samples to form a balanced dataset. Removing data at random leads to losing critical data points in the majority class and several works propose methods for cleverly choosing samples that can be removed without losing important information about majority classes (Lin et al., 2017b; Wilson, 1972; Tomek et al., 1976). Wilson (1972) proposes an Edited Nearest Neighbours algorithm (ENN), where majority class data points that do not agree with the predictions of the KNN algorithm are removed. Another method, Cluster Centroids, undersamples by replacing clusters of the majority class, found with a $k$-means algorithm, by their respective centroids (Lin et al., 2017b). These methods are problematic when data is extremely high-dimensional, as nearest-neighbor classifiers tend to become uninformative in this regime (e.g. $\ell_2$ distance is generally not a good measure of similarity between images). Furthermore, undersampling prevents the user from leveraging the abundance of majority class data to learn better feature representations.

**Classifier-level methods** modify the training routine to emphasize the minority class samples. Several disparate techniques exist in this category. For example, cost-sensitive learning works by altering the loss on minority class points either through re-weighting the loss or by altering the learning rate (Elkan, 2001; Kukar et al., 1998; Cui et al., 2019; Lin et al., 2017a). Intuitively, applying different weights to training samples is similar to over-sampling those data points with the appropriate frequencies. Other classifier-level methods include regularizers which promote large margins on minority class data or impose constraints on the "balanced performance" as measured on a small balanced dataset (Sangalli et al., 2021; Huang et al., 2016; Li et al., 2019). Finally, there are post-processing methods designed to re-scale the scores output by a classifier to achieve better performance (Richard & Lippmann, 1991; Chan et al., 2019).

## 2.2 MAML

Meta-learning algorithms, originally designed for few-shot learning, train on a battery of "tasks" (e.g., classification problems with different label spaces) with the goal of learning a common feature extractor that can be quickly fine-tuned for high performance on new tasks using little data.

Model-Agnostic Meta-Learning (MAML), a popular algorithm for few-shot learning, is compatible with any model and can be adapted to a variety of training algorithms (Finn et al., 2017). The MAML pipeline contains an inner and an outer loop. At the beginning of an episode, tasks accompanied by small *support* and *query* datasets are sampled. In the inner loop, the model's initial *meta-parameters* are fine-tuned on support data in order to perform well on the sampled tasks. Then, in the outer loop, the model is evaluated on query data, and its meta-parameters are updated so that the model performs better on the query data after fine-tuning on support data. The outer loop thus requires unrolling the inner fine-tuning procedure and computing the gradient with respect to the meta-parameters. The resulting meta-parameters have the property that they learn quickly from few-shot data when samples are scarce in a new domain. Our algorithm, MetaBalance, is based on the MAML training routine and exploits the few-shot performance of meta-learning to handle the case where minority-class data is scarce.

Two works, L2RW and Meta-Weight-Net (MW-Net), have applied the MAML algorithm to learn loss weights for training neural networks in the class-imbalance setting (Ren et al., 2018; Shu et al., 2019). Both methods find optimal sample weights by minimizing the meta loss on a balanced holdout set. The main difference between the works is that L2RW learns the sample weights implicitly, while MW-Net trains an explicit weighting function. Our work is different from these algorithms in two aspects: (1) instead of learning the optimal sample weights, we learn the optimal network parameters directly, (2) MetaBalance does not require a balanced holdout dataset, which might be problematic in tasks with few minority samples. Indeed, we show that MetaBalance outperforms MW-Net on tasks with extremely small minority classes and that our algorithm prevents overfitting in these scenarios. Finally, several works target class-imbalance in few-shot learning (FSL). Lee et al. (2019) proposes Bayesian framework for balancing the effect of meta-learning and task-specific learning across tasks and Ochal et al. (2021) demonstrates that re-balancing strategies are still effective against imbalance in FSL.

## 3 METABALANCE

Our algorithm, MetaBalance is based on Model-Agnostic Meta-Learning (MAML), an approach designed for few-shot learning (Finn et al., 2017). Standard meta-learning schemes consist of "inner" and "outer" loops, each with their own, potentially distinct, loss function. In the inner loop, the model is fine-tuned by minimizing a loss function defined by a set of support data. The outer loop evaluates the fine-tuned model on a batch of query data from the same task.

Our proposed MetaBalance method exploits the fact that meta-learning decouples the inner- and outer-loop loss functions, allowing us to use different class balancing strategies for each. We often find that using the natural class (im)balance on the inner loop support set is optimal, as it prevents the optimizer from aggressively overfitting to minority class data. At the same time, we can use re-balanced data in the query set, which guides the algorithm to achieve high accuracy on class-balanced data as training proceeds. This allows us to simultaneously combine the benefits of training on imbalanced data (prevention of over-fitting) with the benefits of training on balanced data (minority classes don't get ignored).

To formalize the MetaBalance algorithm, consider neural network $f$ with parameters $\theta$ that maps samples to predictions. In the inner loop of the MetaBalance training routine, we minimize the loss over sampled batch $X$ to obtain new parameters $\theta' = \theta - \alpha \nabla_\theta \mathcal{L}_X(f_\theta)$. Then, in the outer loop, we evaluate the model on a new balanced sample $Z$ based on new parameters $\theta'$ and compute a loss $\mathcal{L}_Z$ on those predictions as well. We accumulate the outer loss over multiple support and query batches, fine-tuning individually on each support batch, and we finally update $\theta$ to minimize the query loss. See Algorithm 1 for details.

**Sampling strategies in the inner and outer loops:**

---

**Algorithm 1** MetaBalance

---

**Require:** Initial parameter vector, $\theta$, outer learning rate, $\eta$, inner learning rate, $\gamma$.
**for** $epochs = 1, \ldots, epoch$ **do**
    $Loss_Z = 0$
    **for** $step = 1, \ldots, metaStep$ **do**
        Sample unbalanced support batch, X, and balanced query batch, Z
        Evaluate $\nabla_\theta \mathcal{L}_X f(\theta)$
        Compute adapted parameters with gradient descent:
        $\theta' = \theta - \gamma \nabla_\theta \mathcal{L}_X f(\theta)$
        Accumulate the loss over Z:
        $Loss_Z = Loss_Z + \mathcal{L}_Z f(\theta') + \beta \mathcal{L}_X f(\theta)$
    **end for**
    Update $\theta \leftarrow \theta - \eta \nabla_\theta Loss_Z$
**end for**

---

The benefits of MetaBalance come from the fact that we can choose different sampling strategies for sampling support and query data in the inner and outer loops. We can improve the performance beyond baselines, often dramatically, by using complementary sampling strategies for the two losses. For each loss, we consider using the natural class balance, over sampling, undersampling, SMOTE, SVMSmote, ENN, and CC (Chawla et al., 2002; Nguyen et al., 2011; Wilson, 1972; Li et al., 2019). Results are presented and discussed in the following sections.

## 4 EXPERIMENTAL SETUP

We evaluate MetaBalance on four tasks: image classification, loan default prediction, fraud detection, and facial recognition. In this section, we describe the datasets, models, and metrics we utilize in each of our experiments.

### 4.1 DATA

**Image classification:** For image classification experiments, we use the CIFAR-10 dataset, which contains 10 classes with 5,000 images per class in the training set (Krizhevsky et al., 2009). We simulate class-imbalance problems in two ways, each with one majority class containing 5,000 images and nine minority classes containing fewer images. In the "severely imbalanced" scenario, minority classes contain five images per class at train time, while in the "moderately imbalanced" scenario, minority classes contain five to 50 images, selected randomly. The test set used to evaluate the models is balanced with 1000 images per class.

**Loan-default prediction:** We consider a tabular dataset for predicting loan default events with 8,046 data points in the majority class and 1,533 in the minority class.[1] Although this imbalance is not as severe as other datasets we consider, we have found this classification problem to be more difficult than the credit-card fraud prediction problem (see Table 2). We randomly split the data, reserving 80% for training and the remaining 20% for testing.

**Fraud detection:** We use another tabular dataset for predicting fraudulent credit-card transactions (Dal Pozzolo et al., 2014). This dataset contains 284,807 transactions of which only 492 (0.172%) are fraudulent. The released data has been transformed under PCA to avoid releasing the original (possibly private) transaction features. As above, we randomly split the data, reserving 80% for training and the remaining 20% for testing.

**Facial recognition:** Finally, we evaluate our method on the Celeb-A dataset containing images of 10,177 celebrities, out of which 1,000 identities are kept in the test set (Liu et al., 2015). Celeb-A contains labels for identities, gender, and age group as well as other non-sensitive attributes. We artificially unbalance the training set with respect to gender by randomly removing 90% of images from female identities while ensuring that at least 2 images per identity are still present. As we show below, models trained naively on such an imbalanced face dataset exhibit lower test accuracy on

---

[1]https://www.kaggle.com/sarahvch/predicting-who-pays-back-loans

females than on males. Nonetheless, facial recognition systems are known to be resilient to dataset imbalance since they are designed to test on different distributions and, in fact, identities than those in their training data. In this work, we consider the facial identification task rather than the simpler one-to-one facial verification task. We aim to reduce the accuracy gap arising from class-imbalance via MetaBalance. In the next section, we show that even though facial recognition systems are somewhat resistant to imbalance, we are still able to improve their performance in this setting.

## 4.2 MODELS

On CIFAR-10, we train ResNet-18 models using cross-entropy loss (He et al., 2016). On the loan default prediction task and the credit-card fraud detection task, we train feed-forward neural networks with two fully-connected layers and five fully-connected layers, respectively, using binary cross-entropy loss. Finally, for facial recognition, we train a ResNet-18 model with the CosFace head using focal loss designed to emphasize minority samples (Lin et al., 2017a; Wang et al., 2018). For a complete description of hyperparameters, see Appendices A.1, A.2, A.3, and A.4.

## 4.3 METRICS

For tasks with balanced test sets, specifically image classification and facial recognition, we evaluate all models using accuracy. However, accuracy may not be informative in cases where data is imbalanced since it is dominated by the performance on majority classes. Therefore, we use AUC-ROC to report performance of loan default prediction and credit-card fraud detection models. For facial recognition, we evaluate models in the identification (one-to-many classification) setting on test identities that do not overlap with training identities and we report rank-1 accuracy. That is, for each image in the test set (probe photo), we find a nearest test-set-neighbor in the feature space using cosine similarity; the model is correct only if the matched image is a photo of the same identity. We then repeat this procedure for every image in the test set and report average accuracy.

## 4.4 BASELINES

In our experiments, we compare MetaBalance with a variety of techniques that handle class-imbalance. For experiments on CIFAR-10, we compare MetaBalance with random minority over-sampling and random majority under-sampling. In addition, we consider strong data augmentation techniques such as mixup (Zhang et al., 2017), CutMix (Yun et al., 2019), and M2M (Kim et al., 2020) for generating additional samples. We also consider combinations of strong data augmentation techniques with oversampling of minority classes, and we denote such training schemes by OS-mixup and OS-CutMix. Also, we compare our results with re-weighting techniques: Meta-Weight-Net (MWN, Shu et al. (2019)) and loss re-weight (LRwt, Ren et al. (2018)). For the tabular datasets, we consider re-sampling techniques such as random oversampling, undersampling, SMOTE (Chawla et al., 2002), SVMSmote (Han et al., 2005), AllKNN (Tomek et al., 1976), and re-weighting techniques such as Meta-Weight-Net (MWN, Shu et al. (2019)), and loss re-weight (LRwt, Ren et al. (2018)).

# 5 RESULTS

## 5.1 IMAGE CLASSIFICATION

In our experiments on CIFAR-10, we simulate two class-imbalance scenarios. First, in the *severely* imbalanced case, the minority classes contain only 5 training images. Second, the *moderately* imbalanced case is where the minority classes contain a random number of images between 5 and 50. When a ResNet-18 model is trained on such data in a naive way, without any re-sampling or strong data augmentation techniques, it achieves 31.52% and 16.14% accuracy on the moderate and severe imbalance scenarios, respectively. In contrast, a model trained with MetaBalance using naive sampling in the inner loop and random oversampling in the outer loop achieves 40.59% accuracy in the moderate case and 29.88% for the severely imbalanced data, see Table 1.

Interestingly, we find that under-sampling performs better than all the other re-sampling techniques for both imbalance scenarios, while MetaBalance outperforms both the re-sampling and re-weighting methods. See the results provided in Table 1.

Table 1: Comparison of test accuracy (%) on CIFAR-10. 'Naive' indicates training on imbalanced data and OS and US denote oversampling of minority classes and undersampling of majority classes, respectively. We consider two imbalance scenarios: moderate imbalance (Mod. Imb.) and severe imbalance (Sev. Imb.). Meta-B denotes MetaBalance, CMix and OS-CMix denote CutMix and CutMix with oversampling.

| | Method | | | | | | | | | | |
|---|---|---|---|---|---|---|---|---|---|---|---|
| | Naive | OS | US | mixup | CMix | OS-mixup | OS-CMix | MWN | LRwt | M2M | Meta-B |
| Mod. Imb. | 31.52 | 29.62 | 36.53 | 25.84 | 23.98 | 31.04 | 17.92 | 30.31 | 37.17 | 25.58 | **40.59** |
| Sev. Imb. | 16.14 | 14.59 | 23.34 | 12.92 | 12.64 | 12.47 | 10.63 | 13.90 | 17.98 | 13.87 | **29.88** |

**Note 1** *Although we cannot apply AUC on a multi-class classification problem, test accuracy might seem like an unfair metric for comparing models in this setting since there might be a majority-minority accuracy trade-off. To compare these models in a fair manner, we additionally explore threshold adjustment and a Bayesian prior re-weighting in Section 6. In these experiments, we find that MetaBalance consistently outperforms other methods.*

## 5.2 LOAN DEFAULT DETECTION

Loan default prediction is a binary classification problem with positive samples (loan default) constituting 19% of the training data. We train shallow feed-forward neural networks using MetaBalance and compare the results with naive training and baselines discussed in Section 4.

We report performance under two modifications of the MetaBalance routine. The first uses undersampling exclusively in the query batch and the second utilizes the data samplers discussed above in both the inner and outer loops. We call the second method Mixed Strategy MetaBalance or MS-MetaBalance. We try different combinations of sampling strategies in both loops and choose the best one.

We find that MetaBalance outperforms all the methods used for comparison and shows improvements of at least 0.6% AUC-ROC. Moreover, the use of SVMsmote as a sampling technique in the inner loop and undersampling in the outer loop, improves the performance of models trained further. With this variant of our method, we see a 1.3% increase in AUC-ROC compared to the naive training. See Table 2 which compares AUC-ROC for various methods. Appendix Table 7 contains error bars corresponding to experiments from Table 2.

## 5.3 CREDIT-CARD FRAUD DETECTION

Credit-card fraud detection is a tabular binary classification problem with positive samples representing fraudulent transactions and negative samples representing legitimate transactions. Positive samples constitute only 0.172% of all data.

In this setting, we show that MS-MetaBalance outperforms all the re-sampling methods, aside from Meta-Weight-Net (MWN), with at least 0.9% improvement in AUC-ROC. One possible reason that re-weighting techniques perform better than MetaBalance in the credit-card fraud detection is that even though the credit-card fraud dataset is highly imbalanced, unlike the CIFAR-10 and loan default, it contains enough minority samples for naive training to learn meaningful representations and obtain high AUC-ROC. We report averages from five runs each in Table 2. For comparisons to other baselines as well as error bars from our experiments, see Appendices A.3 and A.5.

## 5.4 FACIAL RECOGNITION

For the facial recognition task, we simulate imbalance by removing 90% of female images from Celeb-A. Even though facial recognition systems are already designed to handle a distributional mismatch between training and testing data, MetaBalance still reduces the performance disparity between males and females from 8.43% to 6.09% by boosting performance on females, the minority class. By improving performance on female data, MetaBalance significantly raises overall accuracy. Table 3 contains the full results of these experiments.

Table 2: AUC-ROC of various training algorithms on the credit-card fraud detection and loan default prediction tasks. MetaBal refers to the MetaBalance routine with undersampling in the outer loop and MS-MetaBal indicates the use of the best performing samplers in both loops. Bold figures reflect the row maximium.

| Dataset | Naive | Over-S | Under-S | Smote | SVMSmote | AllKNN | MWN | LRwt | MetaBal | MS-MetaBal |
|---|---|---|---|---|---|---|---|---|---|---|
| | | | | | Sampling Method | | | | | |
| CC Fraud | 0.968 | 0.966 | 0.966 | 0.968 | 0.968 | 0.967 | **0.986** | 0.976 | 0.967 | 0.977 |
| L. Default | 0.655 | 0.647 | 0.645 | 0.616 | 0.633 | 0.660 | 0.659 | 0.658 | 0.666 | **0.668** |

Table 3: Accuracy of face recognition models. We compare models trained in a naive way, trained with oversampling, and trained using our MetaBalance routine.

| Sampling Method | Total Accuracy | Male Accuracy | Female Accuracy |
|---|---|---|---|
| Naive | 90.12% | 95.31% | 86.88% |
| Oversampling | 84.67% | 92.98% | 79.48% |
| MetaBalance | **91.90%** | **95.65%** | **89.56%** |

## 6 METABALANCE PREVENTS OVERFITTING

Overfitting to training data is always a concern of over paramaterized models. When training data is not uniformly distributed across classes, overfitting becomes even more problematic. Models typically overfit disproportionally to minority class data, resulting in high performance gaps between majority and minority classes at inference. We hypothesize that MetaBalance is effective at handling the class imbalance partly because it prevents severe overfitting to scarce minority data. In this section, we analyze the overfitting behavior of a naively trained model, a model trained with oversampling, and a model trained with MetaBalance on the imbalanced version of CIFAR-10.

In Figures 1 and 2, we show that models trained naively or with oversampling overfit on both minority and majority classes. However, the test time accuracy of these models is lower than random on minority classes and close to 100% for the majority class. In fact, the models achieve near perfect accuracy on the majority data in a trivial manner by placing nearly all images into the majority training class. This indicates that that the model does not learn meaningful decision boundaries, even to distiguish points in the majority class from those outside of it. Instead, the model memorizes minority class instances and gerrymanders the decision boundary around them. In contrast, the model trained with MetaBalance achieves only 60% accuracy on the majority class both at train and test time. At the same time, the accuracy on minority classes averages around 30%, a huge increase compared to naively trained models. See Figure 2 for comparisons of training and testing accuracy during training.

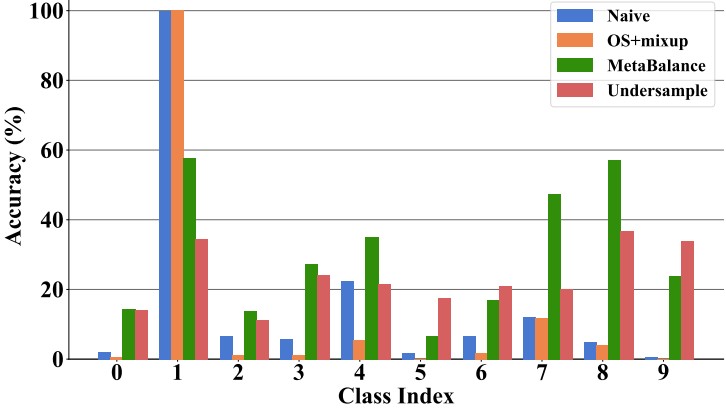

Figure 1: Comparing test accuracy over a balanced test set from models trained on imbalanced data with 5,000 images in the majority class (index 1) and 5-50 images in each of 9 minority classes.

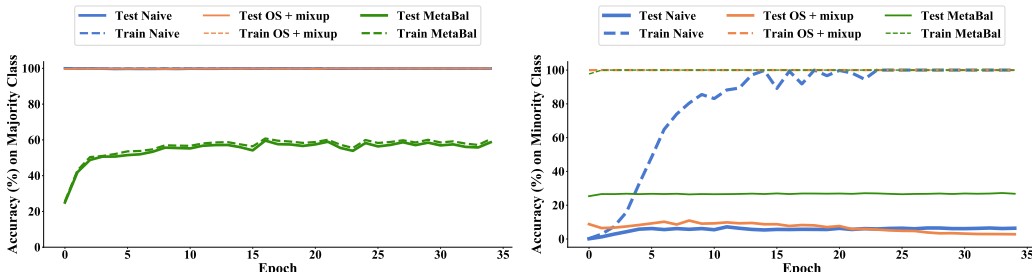

Figure 2: **Left:** majority class accuracy of models trained in a naive way, with oversampling+mixup and with MetaBalance on the severely imbalanced CIFAR-10 dataset. **Right:** average accuracy over minority classes for the same models. Solid lines denote test accuracy and dashed lines denote train accuracy.

One might hypothesize that the naively trained model performs worse on test data primarily because the prior probabilities it learns from training data are highly concentrated on the majority class, while test data is perfectly balanced. Keeping in mind that neural networks trained with softmax cross-entropy loss estimate the posterior distribution over classes, we can improve the performance by adjusting the model's priors to the test distribution. To this end, we divide the confidence scores output by the naively trained model by the frequencies of classes in the training data. We find that re-adjusting the priors indeed improves the test accuracy of the naive model from 16.14% to 22.78% and results in 98.3% accuracy on majority class and 14.3% accuracy on the minority class. However, MetaBalance still achieves superior performance with 29% overall accuracy. This experiment indicates that the naively trained model does not simply learn the wrong prior probabilities over the classes, but it does not actually learn to recognize the patterns contained in minority classes, and it instead blindly classifies inputs into the majority class.

In an effort compare the naive method with MetaBalance on a level playing field, we set the decision threshold of the naive model to match the accuracy of MetaBalance on the majority class. Note that this strategy is not viable in practice and only serves as a object of scientific study, since setting the threshold for the naive method requires knowledge of the full testing set along with its ground-truth labels. We find that in this case, the accuracy of the naive model grows from 16.14% to 27.03%, but it is still lower than the accuracy of the model trained with MetaBalance.

Finally, undersampling causes massive information loss as we must discard the bulk of majority class training data in order the balance the training set. Intuitively, we expect that discarding so much majority class data when data is already not overly abundant is sub-optimal. Thus, while models trained with undersampling perform well on minority classes, they perform very poorly on the majority class, see Figure 1. This phenomenon stands in contrast to naive and oversampling routines which cause models to perform poorly on minority classes. We conclude that a valuable training routine for handling class imbalance should perform better on minority classes than oversampling and naive models while also outperforming undersampling models on majority classes, and it should achieve higher overall test accuracy than these competing methods. In our experiments MetaBalance clears these bars.

## 7    DISCUSSION

A vast body of real-world problems demand that deep learning systems accommodate highly imbalanced training data in order to be useful for practitioners. In this paper, we introduce a new method to handle class-imbalance, motivated by meta-learning. Specifically, we exploit the benefits of having an inner and outer loop, so that we are consequently able to decouple the sampling strategies in the two loops. We demonstrate that our method, MetaBalance, improves performance over existing methods across various datasets from diverse domains when the number of samples in minority classes is small. While investigating the reasons behind this method's success, we observe that it prevents overfitting to majority classes.

## 8 ETHICS STATEMENT

While MetaBalance performs well in our experiments, practitioners should be cautious since real-world datasets are heterogeneous, and their own datasets might not reflect those in our experiments. Even though MetaBalance appears to outperform existing methods in most experiments, we note that class imbalance problems can be extremely difficult, and even the performance of our own method may not satisfy deployment expectations in some critical use-cases.

## 9 REPRODUCIBILITY STATEMENT

The code for all our experiments can be found in the supplementary material and the implementation details of each experiment can be found in the Appendix. We use publicly available datasets and reference them in the paper, so readers can easily download the data and reproduce our experiments. For the M2M (Kim et al., 2020) approach we use the authors' implementation included in their paper.

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

# A  APPENDIX

## A.1  IMAGE CLASSIFICATION

For the resampling techniques, we train a ResNet-18 with Nesterov accelerated SGD and a learning rate of 0.01, momentum of 0.9, and weight decay of 0.0005. We train the neural network for 350 epochs with a cosine annealing scheduler and a batch size of 20. For the meta-weight-net, since the validation dataset needs to be balanced and disjoint, we were constrained by the minimum number of images in all the classes. Hence we created a validation dataset with different sizes (each class has 1, 2 or 3 images in the validation dataset) and used an Nesterov accelerated SGD with a momentum of 0.9, weight decay of 0.0005 and a range of learning rates (0.01, 0.02 and 0.05). We find the meta-weight-net produced best results when learning rate is 0.02 and the number of images from each class is 1 in the validation dataset for the severe class imbalance case, while for the moderate imbalance we see the best numbers are for learning rate of 0.05 and when the number of images in the validation dataset is 1. For the case of M2M we use the hyperparameters provided in the paper to train our model on both the types of class imbalance. For the case of loss re-weight, we do a hyperparameter search on different values of $\beta$ (0.9999, 0.999, 0.99, 0.9) and different values of learning rates (0.01, 0.02 and 0.05) and report numbers on best combination which were 0.99 and 0.02 respectively for both the severe and moderate class imbalance. For MetaBalance, $\theta'$ is calculated as described in 1 with a constant $\gamma$ of 0.01, and the losses are accumulated with a constant $\beta$ of 0. The losses are accumulated over 80 meta steps before updating $\theta$. We use support batch size 20 and query batch size 30.

## A.2  LOAN DEFAULT DETECTION

For this task, we train a two-layer feed-forward network. The first layer takes in an input of length 12 and outputs 25 features, while the second layer is the output layer that predicts loan default. We use a ReLU activation function. For the re-sampling techniques discussed in paper (naive, over-sampling, under-sampling, smote, svmsmote, allknn, borderliesmote, adasyn, nearmiss, clustercentroids, smoteenn) , we employ the Nesterov accelerated SGD optimizer, with a momentum of 0.9 and weight decay of 0.0005 and a range of learning rates (0.01, 0.02, 0.05 and 0.1), and we train networks for 100 epochs and report the best results over different learning rates. The learning rate that produces the best results for the methods discussed above are in the table 4. Again, we split the original dataset into training and testing data with an 80%/20% split and we use a batch size of 24. For the meta-weight-net, we use Nesterov accelerated SGD optimizer, with a momentum of 0.9 and weight decay of 0.0005 and do a hyperparameter search over different learning rates (0.01, 0.02, 0.05 and 0.1) and different sizes of balanced validation dataset (1%, 2%, 5% and 10%) and report the best results. We find that the best results for meta-weight-net were produced with a learning rate of 0.01 and when 1% of the training data is in the validation dataset. Similarly for loss re-weighting strategy we use SGD optimizers, with 0.9 momentum and 0.0005 weight decay and do a hyperparameter search on different values of $\beta$ (0.9999, 0.999, 0.99, 0.9) and different learning rates (0.01, 0.02, 0.05 and 0.1) and report results on best combination which were 0.999 and 0.1 respectively. For MetaBalance, we train with the Nesterov accelerated SGD optimizer, with a learning rate of 0.02, momentum of 0.9 and weight decay of 0.0005. $\theta'$ is calculated as described in 1 with a constant $\gamma$ of 0.01 and the losses are accumulated with $\beta = 0.01$. The losses are accumulated until the 80 meta steps before updating $\theta$. We use support batch size 24 and query batch size 16.

## A.3  CREDIT-CARD FRAUD DETECTION

For credit-cart fraud detection, we train a five-layer feed-forward network, and the numbers of channels in each layer is shown in Table 5. This model has 29 input features and hidden layers of sizes 16, 24, 20, and 24 followed by an binary classification output that predicts fraud. We also use dropout after the second layer with a probability of 0.5. After each layer (other than the last) there is a ReLU nonlinarity. For the re-sampling techniques, we train for 100 epochs and use the SGD optimizer with Nesterov momentum with a coffecient of 0.9, a weight decay coefficient of 5e-2 and a range of learning rates (0.01, 0.02, 0.05 and 0.1) out of which we report the numbers with best results. The learning rates that produce the best results for each of the re-sampling techniques is given in the table 4. We split the original dataset into training and testing data with an 80%/20% split and we use a batch size of 24. For meta-weight-net we take the training set created in the same way as before

Table 4: Learning rates that produces the best results for each of the popular re-sampling techniques

| | Dataset | |
| --- | --- | --- |
| Sampling Method | CC Fraud | Loan Default |
| Simple | 0.1 | 0.05 |
| SMOTE | 0.1 | 0.1 |
| BorderlineSMOTE | 0.1 | 0.1 |
| SVMSmote | 0.1 | 0.1 |
| ADASYN | 0.1 | 0.1 |
| Over-sampling | 0.05 | 0.1 |
| Cluster-Centroids | 0.01 | 0.01 |
| Under-sampling | 0.1 | 0.1 |
| NearMiss | 0.05 | 0.05 |
| AllKNN | 0.1 | 0.05 |
| SMOTEENN | 0.02 | 0.05 |

and create a balanced validation dataset by removing different percentages (1%, 5%, 10% and 20%) of data, and for each combination we use SGD same as previous setting with a range of learning rates (0.01, 0.02, 0.05 and 0.1) and finally report the best results which we get when we use a learning rate of 0.02 and use 20% of the data for validation. For the loss re-weighting strategy we use a range of $\beta$ (0.9999, 0.999, 0.99 and 0.9) and a range of learning rates (0.01, 0.02, 0.05 and 0.1) and report results on best combination which were 0.9 and 0.01 respectively. For MetaBalance, we use SGD with Nesterov momentum with a coefficient of 0.9, a weight decay coefficient of 5e-2, and an initial learning tare of 0.01. $\theta'$ is calculated as described in Algorithm 1 with a constant $\gamma$ of 0.01, and the losses are accumulated with a constant $\beta$ of 0. The losses are accumulated until 80 meta steps before updating $\theta$. We use support batch size 24 and query batch size 16.

Table 5: The number of channels in each layer of the feed-forward neural networks used for credit-card fraud detection.

| Input Size | First Layer | Second Layer | Third Layer | Fourth Layer | Output Layer |
| --- | --- | --- | --- | --- | --- |
| 29 | 16 | 24 | 20 | 24 | 1 |

Table 6: AUC-ROC for other training algorithms on the credit-card fraud detection and loan default prediction tasks.

| | Sampling Method | | | | |
| --- | --- | --- | --- | --- | --- |
| Dataset | BorderlineSMOTE | ADASYN | NearMiss | ClusterCentroids | SMOTEENN |
| CC Fraud | 0.950 | 0.963 | 0.926 | 0.966 | 0.955 |
| Loan Default | 0.632 | 0.612 | 0.604 | 0.558 | 0.617 |

## A.4 FACIAL RECOGNITION

We train a Resnet-18 with SGD, a learning rate of 0.1, momentum of 0.9, and weight decay of 0.0005 computed only on parameters without batch normalization. We train the neural network for 120 epochs with a multi step scheduler (stages - 35, 65 and 95) and a batch size of 128. For MetaBalance, we train the model with same optimizer as in the above setting but with learning rate drops at stages 50 and 100. We use support batch size 128 and query batch size 128, formed by concatenating 2 batches each of size 64 sampled separately from both genders. $\theta'$ is calculated as described in 1 with a constant $\gamma$ of 0.01 and the losses are accumulated with a constant $\beta$ of 0. The losses are accumulated over 4 meta steps before updating $\theta$.

## A.5    ERROR BARS

Table 7 shows the standard error over 10 runs of each technique on the fraud and loan datasets. Table 7 shows the standard error over 4 runs of different techniques on the Cifar 10 class imbalance experiments.

Table 7: Standard error of AUC-ROC for credit card fraud detection

| Dataset | Naive | Over-S | Under-S | Smote | SVMSmote | AllKNN | CC | MetaBal | MS-MetaBal |
|---|---|---|---|---|---|---|---|---|---|
| CC Fraud | 0.006 | 0.003 | 0.003 | 0.008 | 0.003 | 0.003 | 0.004 | 0.004 | 0.002 |
| Loan Default | 0.009 | 0.013 | 0.004 | 0.008 | 0.011 | 0.002 | 0.009 | 0.003 | 0.004 |

Table 8: The Standard Errors of Test Accuracy for the Cifar 10 Experiments

| | Naive | OS | US | mixup | CMix | OS-mixup | OS-CMix | MWN | LRwt | M2M | Meta-B |
|---|---|---|---|---|---|---|---|---|---|---|---|
| Mod. Imb. | 0.806 | 0.837 | 0.096 | 0.523 | 0.299 | 0.471 | 0.352 | 0.826 | 2.00 | 0.366 | 0.321 |
| Sev. Imb. | 0.238 | 0.358 | 0.309 | 0.109 | 0.095 | 0.340 | 0.070 | 0.953 | 0.564 | 0.622 | 0.165 |

## A.6    COMPUTE RESOURCES

In order to train using MetaBalance, for both severe and moderate class imbalance on CIFAR-10, we require approximately 100 Nvidia GeForece RTX 2080Ti GPU hours. For both credit card fraud and loan detection we require less than 2 hours of 2080Ti compute time. For facial recognition, we require 52 2080Ti GPU hours.

