# OpenReview forum: "MetaBalance: High-Performance Neural Networks for Class-Imbalanced Data"
_ICLR.cc/2022/Conference — ICLR 2022 Submitted_

### Official Review · Reviewer_5Nup · 2021-10-26

**Correctness:** 3
**Technical Novelty And Significance:** 2
**Empirical Novelty And Significance:** 3
**Recommendation:** 5
**Confidence:** 4

**Main Review:**

This paper aims at alleviating the class-imbalance problem, which is indeed a common and important problem in many real-world scenarios. The authors have proposed a novel strategy and gave detailed introductions of previous works to clarify their differences. Generally, the proposed algorithm is easy to implement and potentially can be easily used for a wide range of applications.

However, the proposed method is not well motivated. The authors have claimed that the method is proposed “to simultaneously combine the benefits of training on imbalanced data (prevention of over-fitting) with the benefits of training on balanced data (minority class don’t get ignored).” This is not enough to illustrate the rationale behind adopting the meta-learning process. Will directly adopting two loss functions (i.e., on imbalanced and balanced data) simultaneously during training provides the same effect? Also, the authors gave some confused explanations of the method design. For example, “we often find that using the natural class (im)balance on the inner loop support set is optimal, as it prevents the optimizer from aggressively overfitting to minority class data.” It is still unclear why there is a need for an inner loop for training with imbalanced data.

Another concern is that the experiments are not sound. The true practical abilities of MetaBalance are hard to assess as the experimental settings are not the typical benchmark settings for previous studies. Specifically, we can observe that MetaBalance outperforms other methods on one real-world imbalanced dataset while worse than the other. On the image classification and facial recognition tasks, the imbalance settings are handcrafted by the authors. Especially, on the image classification task, the authors only built two settings (i.e., moderately imbalanced and severely imbalanced). Detailed experiments on how the model performance changes with the degree of imbalance (or the scale of minority class) are not provided. Then, it is unclear if MetaBalance steadily improves the model performance when training with different degrees of class imbalance.

The last weak point is that the authors did not give any theoretical analysis on their method, which can provide more insights into the traits of their method and better prove its effectiveness.

Based on the above concerns, I am not fully convinced about the proposed method.


**Summary Of The Paper:**

This paper introduces a novel MAML-based learning strategy to alleviate the class-imbalance problem, namely MetaBalance. The authors have conducted experiments on four datasets to show how their proposed method can improve classification performance when the model is trained with severely imbalanced data.

**Summary Of The Review:**

While this paper studies a practical and important problem, the technical parts are not well-motivated and experiments are not very sound.

---

### Official Review · Reviewer_tbZ3 · 2021-10-29

**Correctness:** 3
**Technical Novelty And Significance:** 2
**Empirical Novelty And Significance:** 2
**Recommendation:** 3
**Confidence:** 5

**Main Review:**

# Strengths
+ The method is simple and straightforward.
+ The evaluation is relatively thorough, spanning across different data modalities where data imbalance is common.


# Main Weaknesses
1. __Limited novelty w.r.t. the literature__
- The novelty is limited. In fact, using meta-learning for imbalanced classification has already been studied in the literature [1]. What is the difference between this paper compared to [1]? Further, to me it seems the whole MetaBalance method is a direct adaptation of MAML, with importance weighting added. This is again with limited novelty, and also raises the question why meta-learning should be the right choice.

2. __Failed to compare with more advanced & related imbalanced learning methods__
- The main weaknesses of the paper is that, it only compares to simple imbalanced learning methods, but failed to compare to advanced methods. In fact, there are bunch of works using other advanced designs with better results on imbalanced classification (e.g., two-stage training [5], ensemble [3], two-stream network architecture [2], or self-supervision [4]). The authors should directly compare to these methods for a complete evaluation to show advantages of MetaBalance, or alternatively (at least) show MetaBalance is complementary to these methods. Otherwise, the improvements are unjustified.

3. __The evaluation protocol is not rigorous enough__
- Although the current evaluation spans across different data modalities, the dataset size is relatively small and might not reflect the real-world cases. For visual recognition tasks, large-scale benchmark datasets for evaluating class-imbalanced learning algorithms should also be considered [3,4,5] (e.g., ImageNet-LT, iNaturalist, etc.). Otherwise, the claim of performance gains is not well justified.

4. __Unclear source of improvements & weird performance drop of the head classes__
- It seems the improvements of MetaBalance come from the sacrifice of the head class performance. Sometimes the performance drop of the head classes are large (Fig. 2). What is the insight behind these observations, and in what scenatios are MetaBalance more useful?

# Other comments / questions
- In Fig.1, although I can see MetaBalance prevent overfitting to the majority class, it seems to underfit all classes (the drop in major class is huge). Is there any explanation on this phenomenon?
- What is the actual computational cost (training time) compared to, say simple baselines like re-weighting/re-sampling methods?

# References
[1] Rethinking class-balanced methods for long-tailed visual recognition from a domain adaptation perspective. CVPR 2020.

[2] Bbn: Bilateral-branch network with cumulative learning for long-tailed visual recognition. CVPR 2020.

[3] Long-Tailed Recognition by Routing Diverse Distribution-Aware Experts. ICLR 2021.

[4] Rethinking the Value of Labels for Improving Class-Imbalanced Learning. NeurIPS 2020.

[5] Decoupling representation and classifier for long-tailed recognition. ICLR 2020.

**Summary Of The Paper:**

The paper proposed to use meta-learning to improve class-imbalanced learning. In particular, it named the method as MetaBalance, and tested it over several data modalities, where data imbalance is common. The empirical results demonstrated the improvements over several previous methods for imbalanced data.

**Summary Of The Review:**

Overall, the problem is important and interesting, however the current draft is not well-justified on the actual algorithm design, the novelty, and (perhaps the most important) empirical comparisons to existing imbalanced learning methods. Further comments/questions are listed in the weaknesses / questions part.

The paper has its potential to the field, but issues need to be addressed first. I'm happy to change my score if the feedback addresses my concerns. Please refer to the points in the weaknesses / questions part. I would like to see feedbacks on these comments/questions.

---

### Official Review · Reviewer_16HT · 2021-10-31

**Correctness:** 2
**Technical Novelty And Significance:** 2
**Empirical Novelty And Significance:** 1
**Recommendation:** 3
**Confidence:** 5

**Main Review:**

Strength
1. The proposed method is rather straightforward extension of MAML, but it is indeed reasonable extension.
1. The proposed method is evaluated not only on an image classification task but also on tabular type datasets.

Weakness
1. The proposed method is rather straightforward application of MAML to the class-imbalanced problem. The key idea when bringing MAML’s idea to class-imbalanced problem is to change the sampling strategy in what is called inner loop and outer loop. However, the idea of using different sampling strategies to deal with class-imbalance problem, though implemented in a different way, is already studied in [A1], which is not discussed in the paper. Given these prior works, the technical novelty of the presented paper is limited.
1. The proposed method should be evaluated on large-scale datasets, too. For example, CIFAR100-LT, ImageNet-LT, Places-LT, and iNaturalist datasets are commonly used.
1. The methods compared with the proposed method are not sufficient as baseline methods. There are a lot of recent methods designed for class-imbalance problem such as [A1-A7]. The proposed method should be compared with these recent methods to show its effectiveness. Especially, the baseline methods on facial recognition dataset is only very simple ones.
1. The experimental protocol of CIFAR dataset does not follow a standard one proposed in the work of [A8]. If there is any special reason to stick to the current protocol, the reason must be clearly stated.
1. Figure 1 and Figure 2 is not sufficient for claiming the strength of the proposed method against overfitting.
```
In Figures 1 and 2, we show that models trained naively or with oversampling overfit on both minority and majority classes. However, the test time accuracy of these models is lower than random on minority classes and close to 100% for the majority class.
```
It is good to show that the proposed method is less prone to overfitting compared with a naïve model and an oversampling model, but it is not enough. An oversampling model is well-known to be prone to overfitting. It is not appropriate to claim that the proposed method has strength against overfitting just because it is better than such weak baselines. It is better to show the advantage of the proposed method in comparison with other methods designed for long-tail recognition such as [A1-A7].

[A1] Kang+, Decoupling Representation and Classifier for Long-Tailed Recognition., ICLR 2020\
[A2] Tan+, Equalization loss for long-tailed object recognition. CVPR 2020\
[A3] Sinha+, Class-Wise Difficulty-Balanced Loss for Solving Class-Imbalance, ACCV 2020\
[A4] Ren+, Balanced Meta-Softmax for Long-Tailed Visual Recognition, NeurIPS 2020\
[A5] Tang+, Long-Tailed Classification by Keeping the Good and Removing the Bad Momentum Causal Effect, NeurIPS 2020\
[A6] Wang+, Long-tailed Recognition by Routing Diverse Distribution-Aware Experts., ICLR 2021\
[A7] Li+, MetaSAug: Meta Semantic Augmentation for Long-Tailed Visual Recognition, CVPR 2021\
[A8] Cui+, Class-balanced loss based on effective number of samples., CVPR 2019


**Summary Of The Paper:**

This paper proposes to apply meta-learning framework presented by Finn et al. to class-imbalanced problems. It is basically simple adaptation, but one twist presented in this paper to make the meta-learning work well is to change the sampling strategy in what is called inner loop and outer loop. The presented method is evaluated not only on an image classification task but also tabular type datasets as well.

**Summary Of The Review:**

The idea of utilizing MAML’s meta learning framework in class-imbalanced problems itself is interesting and good direction. However, the technical novelty of the paper is only marginal because it is rather straightforward application of MAML. The empirical novelty is not enough, too. It is good to try to evaluate the method on various tasks other than image classification tasks, but the proposed method should be compared with recently proposed methods in a fair protocol to show the advantage of the proposed method over these prior works.

---

### Official Review · Reviewer_bejS · 2021-11-01

**Correctness:** 3
**Technical Novelty And Significance:** 3
**Empirical Novelty And Significance:** 3
**Recommendation:** 6
**Confidence:** 4

**Main Review:**

The idea of using meta-learning to handle class imbalance is quite interesting. The paper is well-organized, covers most of the literature, and presented results show the potential of the method. However, there are still several points that are crucial:

- To my understanding, the query set is always balanced which means that the performance is optimised for a balanced test set. However, in many practical applications, test and train sets have similar degrees of skewness. So, I think it would be nice to present results on imbalanced test sets as well. Recent works like the one below present results for both balanced and imbalanced cases which make sense:
https://arxiv.org/pdf/2102.12894.pdf

- Recent methods that propose new loss functions are not included to the comparisons e.g. the ones proposed by Sangalli et al. and Li et al. I think it is crucial to compare with these methods to better position the proposed method in the recent literature.

- In the data section, the paper only mentions train and test splits but not the validation splits. How are the models monitored during training without validation sets? For example, using validation set to determine hyper-parameters (beta, metaStep etc.) is quite crucial.  Also, details about hyper-parameter selection e.g. range of values that have been tried should be explicitly mentioned.

- For the binary classification experiments, FPR at certain TPR is another metric that is commonly used in the literature. I think results with additional methods would be beneficial.

- In the multi-class classification experiments, there is only a single majority class and multiple minority classes. How would the performance change if there were a single minority class and multiple majority classes? I think this is a more challenging problem since the impact of minority class will be even less and it is more consistent with the experiment in the literature.


**Summary Of The Paper:**

The paper proposes a meta-learning-based method to learn under class imbalance. Class imbalance is ubiquitous in many real applications and neural networks tend to learn biased models toward majority classes. Analogous to a popular meta-learning method called MAML, the proposed method consists of 2 nested training loops: outer on the query dataset and inner on the support dataset. In the inner loop, support datasets are randomly sampled from the original dataset by retaining the class imbalance and candidate parameters are obtained by updating the current parameters with the gradient of the loss on the support dataset. During the inner loop, a global loss on both the query and the current support datasets is accumulated.  The model parameters are updated after running the inner loop for a pre-defined number of steps using the gradient of the accumulated loss. By doing so, it is expected to learn a model from different subsamples (support set) of the imbalanced datasets while performing well on a balanced dataset.
The proposed method is compared with several undersampling/oversampling/re-weighting based methods to handle class imbalance. The presented results show that MetaBalance improve accuracy on minority classes in many cases and avoid overfitting to minority/majority class samples.

**Summary Of The Review:**

To sum up, I am quite positive about the paper and I think it mostly satisfies high publication standards for ICLR. However, there are still major things which may be added during revision:

- Comparison with more recent methods (Sangalli et al. and Li et al.). Sangalli et al. is more recent and compares with the methods in Li et al. So, I think it would be sufficient to compare with the former one.
- Hyperparameter selection: I think this is the most crucial part. I wonder how this is performed without validation set. This should be clarified and all details about hyper-parameter selection should be given.
- Using additional metrics where possible would be beneficial.

---

### Decision · Program_Chairs · 2022-01-20

**Decision:**

Reject

**Comment:**

This paper proposes a method for class-imbalanced data based on meta-learning. The technical contribution of the proposed method is limited as it is a reasonable but straightforward extension of the existing method. In addition, as commented by the reviewers,
the comparison with existing methods is not enough, it is unclear why it is meta-learned with balanced test data, and hyperparameter tuning details are not given.